# Standalone Valuation Method for Software-as-a-Service Operational Knowledge Derived from Human Intellectual Capital Qualitative Changes

Suguru Sakuma * and Tomoyuki Furutani

Faculty of Policy Management, Graduate School of Media and Governance, Keio University,
Fujisawa-shi 252-0882, Kanagawa, Japan; maunz@sfc.keio.ac.jp
* Correspondence: sasusa@sfc.keio.ac.jp; Tel.: +81-466-49-3623

**Abstract:** This study focuses on digital operational knowledge belonging to natural persons and proposes a greenfield approach to differentiate the value of intangibles from that of human intellectual capital. Our research approach involves two assessments. Assessment 1 evaluates intangible assets using the internally generated goodwill (IGG) measure. We analyze time-series IGG data for six digital sectors, using the top 90 software-as-a-service (SaaS) companies as a benchmark. The results indicate that the IGG of the SaaS benchmark is higher than the total IGG of the six sectors. Assessment 2 focuses on the correlation between digital labor investment and digital investment returns before and after 2013 for the six sectors to identify positive and negative correlations from 2013 onward. The results indicate that, since 2013, a qualitative change has occurred in digital labor capital that has not been reflected in financial statements because of accounting distortions and that the returns on investment for digital labor have been underestimated. The standalone valuation of digital know-how that belongs to natural persons, previously based on operating expense, will be based on capital expenditure. In addition, amortization will have the same contribution as depreciation of tangible assets to value creation.

**Keywords:** SaaS; digital intangible; value transfer; human intellectual capital; accounting distortion; digital labor productivity

## 1. Introduction

The global digital revolution led to the rise of diverse digital assets, notably regarding the assessment of employee digital literacy, also referred to as digital operational proficiency. Digital intangibles present challenges to current accounting regulations as they are often difficult to identify as sources of revenue, leading to significant accounting discrepancies (Visconti 2020). The valuation of such human capital intangibles as assets of income is a slippery and complex process that results in their unavailability for recording on a balance sheet (BS) (Qin 2017).

In this study, we examined the human capital value of employees of software-as-a-service (SaaS) companies and developed a quantifiable valuation method for these intangible assets. The debate on digital tangibles is important because the relevance and reliability of intangible assets in accounting and financial reports have been questioned. The literature on the consequences of a lack of accounting recognition of intangibles has been critically reviewed (Zéghal and Maaloul 2011), but the relationship between accounting choices associated with intangibles and their value relevance, and the mitigating effects of firm lifecycles, must be considered (Jaafar 2011). Past initiatives, in contrast with contemporary practices, have provided solutions for accounting for intangible assets (Lai 2011). However, there is continuing debate on how to represent digital intangible valuations in accounting. Value relevance depends on how well accounting captures firm digital intangibles, how they are valued, and the trustworthiness of the assessed values (Saunders and Brynjolfsson 2016).

The current concept of labor productivity does not accurately reflect the value of SaaS digital operational knowledge. Therefore, this study proposes redefining "digital labor productivity" and reclassifying "labor investment" as "digital labor investment" and "capital investment" as "digital capital investment". We emphasize the importance of disclosing the quantifiable value of an employee's knowledge, experience, creativity, and expertise, which form the human capital assets of a company. Our goal is to legitimize the recognition of SaaS operational knowledge as an intangible asset on a BS through sound analytical arguments. Additionally, if digital intangibles are recognized and listed on a BS, they can provide a foundation for the implementation of the basic income concepts.

Regarding basic income concepts, the value of digital operations lies in human capital. DevOps is used as a quantitative index of human capitalization to leverage advanced digital operations as a source of revenue. DevOps incorporates various operational methodologies with the concept of continuous integration/continuous deployment (CI/CD) and represents a novel structuring method that treats digital operational knowledge as a new revenue source (Badshah et al. 2020).

NFTs (non-fungible tokens) and cryptocurrencies have recently materialized, and a market for trading them has been formed. However, since it is impossible to materialize digital operation capabilities, companies have been forced to internalize the digital operation value of their employees, giving rise to the idea of CI/CD. The prevailing mainstream agile method evolved into development forms such as DevOps and AIOps, which integrate development and operations and contain circularity properties.

The remainder of this paper is organized as follows. Section 2 presents the literature review; Section 3 details the research approach adopted in this study; Section 4 presents the results of two different types of assessments; Section 5 discusses additional considerations based on the results presented in the previous sections; and Section 6 summarizes our conclusions.

## 2. Literature Review

### 2.1. What Is Accounting Distortion?

The treatment of tangible and intangible investments in current accounting standards can lead to financial metric distortions and misrepresentation of a company's value. The intangible investments of digital companies are excluded from financial reports because they cannot be capitalized, and research indicates a difference in value relevance between tangible and intangible investments. Accounting distortions result from accrual accounting and may be caused by accounting standards, estimation errors, and conservatism. Conservatism tends to produce a pessimistic bias in financial statements.

### 2.2. Relationship between Digital Intangibles and Human Capital

The digital revolution has introduced new intangible assets that are difficult to value accurately, including NFTs and virtual currency creditworthiness. Digital intangibles related to human intellectual capital, such as operational knowledge, are challenging to quantify in financial terms. This has led to accounting distortions, such as base erosion and profit shifting (BEPS), based on the absence of a social consensus regarding how to record these intangibles as human intellectual capital. The Organization for Economic Co-operation and Development (OECD) BEPS Conference of 2018 defines digital intangibles as hard-to-value intangibles (HTVI) (OECD 2018). BEPS occurs when companies shift profits to low-tax jurisdictions, resulting in revenue losses in high-tax countries.

### 2.3. Various Approaches to Digital Intangibles Valuation

Traditional valuation methods such as the discounted cash flow method are no longer adequate for valuing digitally advanced companies, leading to the development of various new approaches. One such approach is the greenfield method of intangible asset valuation based on the IVS 210 standard, which can capture the value of standalone human capital and accommodate dynamic changes in digital assets (Clohessy et al. 2020). Measurement

methods for SaaS platform contributions to GDP creation and productivity improvement for self-employed individuals have been suggested (Ahmad and Schreyer 2016). These include statistical measurement methods (Raghavan et al. 2020; Labaye and Remes 2015; Simonsson and Magnusson 2018), empirical analyses of digital value chain innovation (Qu et al. 2017), brand value platform creation (Sriram et al. 2006), examination of the impact of the technology paradox on productivity (Arbia et al. 2019), and correlations between spatial economic models and economic growth. Demand forecasting modeling (Kourentzes and Petropoulos 2016) and virtual currency valuation modeling are among the proposed pricing methods (Hunter and Kerr 2019).

National empirical case studies have been conducted in Germany, focusing on the productivity factors of internal and external collaboration innovation (Hensen and Dong 2020), and in Canada, examining the relationship between digital training and economic growth (Walker et al. 2018). Multi-factor productivity studies using sector-level data from the European Union have also been conducted (Emvalomatis 2017), and digital innovation valuation in Africa has been analyzed using a sector-based approach (Foster et al. 2018). Productivity growth across many developed economies has been analyzed using a multi-country and multi-sector approach (Remes et al. 2018). In this study, we adopted the sector-based method proposed for valuing digital intangibles in Japan, which provides insights into evaluating the value of digital intangible assets in the Japanese market accurately.

## 3. Materials and Methods

### 3.1. Research Approach

Our research approach involved two assessments. Assessment 1 examined the value transfer phenomenon across digital sectors in the Japanese market. The SaaS bubble in 2020 was used as a baseline for the greenfield method, which calculates the standalone value of each sector. To isolate the value of intangible assets, internally generated goodwill (IGG) was used as an evaluation index for intangible assets, and time-series data on IGG differences were calculated for six digital sectors containing 90 SaaS benchmark companies. $\beta$-correction was performed for each sector to correct the accounting distortion caused by human capital. Assessment 2 examined the correlation between digital labor investment and investment returns in six sectors before and after 2013 to identify sectors with positive and negative correlations. The goal of this assessment was to verify the consistency of the intangible asset inflow and outflow sectors with respect to the correlation between digital labor investment and investment returns. Inconsistent sectors indicate that the accounting distortions might be caused by qualitative changes in digital labor capital. The proposed valuation method considers the human capital factor of digital operations when valuating digital intangible assets.

### 3.2. Assessment 1: Value Transfer between SaaS Sectors in the Japanese Digital Market

#### 3.2.1. Purpose

We define the price-to-book ratio (PBR) multiples to represent IGG. Using the top 90 SaaS listed companies in the Japanese market in 2020 as a benchmark, IGG is split by digital sector to calculate the IGG differences between sectors. The IGG values and PBR multiples represent the gross valuation of digital intangibles. The historical transition of IGG is verified against the benchmark.

#### 3.2.2. Extraction Rule for SaaS-Advancing Companies

We define SaaS-advancing companies as those who (1) provide their own subscription-type service, (2) are included in the top 90 SaaS companies in terms of sales in the Japanese market in 2020, and (3) are listed on the Tokyo Stock Exchange.

### 3.2.3. Digital Sectors to Which the 90 SaaS Companies Belong

The six sectors to which the top 90 SaaS companies belong are IT infrastructure services (N = 41 companies), business process outsourcing (BPO) services (N = 58 companies), system development (N = 195 companies), software services (N = 195 companies), specialized information media (N = 166 companies), and media advertising services (N = 41 companies) (Amiri 2022). The details of these sectors are listed in Table 1.

**Table 1.** Ratio of SaaS (Upper%: intra-sector ratio; lower%: ratio within benchmark).

| Digital Sector | #SaaS@2020 | | $p$ (F $\leq$ f) | $p$ (T $\leq$ t) | #Non-SaaS@2020 | | $p$ (F $\leq$ f) | $p$ (T $\leq$ t) | #Total |
|---|---|---|---|---|---|---|---|---|---|
| Infrastructure Services | 12 | 29.3%<br>13.3% | 0.093 ‡ | 0.483 [†1] | 29 | 70.7%<br>4.8% | 0.471 ‡ | 0.483 [†7] | 41 |
| BPO Services | 3 | 5.2%<br>3.3% | 0.000 ‡ | 0.000 [†2] | 55 | 94.8%<br>9.1% | 0.396 ‡ | 0.436 [†8] | 58 |
| System Development | 16 | 8.2%<br>17.8% | 0.034 ‡ | 0.001 [†3] | 179 | 91.8%<br>29.5% | 0.422 ‡ | 0.440 [†9] | 195 |
| Software Services | 43 | 22%<br>47.8% | 0.011 ‡ | 0.000 [†4] | 152 | 78%<br>25.1% | 0.221 ‡ | 0.279 [†10] | 195 |
| Specialized Information Media | 9 | 5.4%<br>10% | 0.000 ‡ | 0.000 [†5] | 157 | 94.6%<br>25.9% | 0.357 ‡ | 0.305 [†11] | 166 |
| Media Advertising Services | 7 | 17%<br>7.8% | 0.074 ‡ | 0.065 [†6] | 34 | 83%<br>5.6% | 0.478 ‡ | 0.409 [†12] | 41 |
| #Total | 90 | 100% | | | 606 | 100% | | | 696 |

‡ f = 0.05, [†1] t = 0.041, [†2] t = 0.160, [†3] t = −3.72, [†4] t = −4.13, [†5] t = −9.34, [†6] t = −0.22, [†7] t = −0.04, [†8] t = 0.16, [†9] t = 0.15, [†10] t = 0.58, [†11] t = 0.51, and [†12] t = −0.22.

To check the statistical significance of each of the six sectors for the SaaS 90 company population and the non-SaaS population, F-test and t-test for each sector are conducted and the statistical significance of the variance and mean are examined. As shown in Table 1, the SaaS population for most of all sectors is statistically significant, while the non-SaaS population is not significant. In other words, the SaaS population is heterogeneous in variance and mean, while the non-SaaS population is homogeneous. Furthermore, since the population of 90 SaaS companies is extremely heterogeneous, it suggests that there might be critical factors that cause this kind of strong heterogeneity.

### 3.2.4. Setting a Benchmark

The greenfield method seeks differential data between each digital sector and the benchmark, as shown in Figure 1, and then analyzes the time series of the SaaS sectors to which the companies belong as well as the accompanying value transfer. PBR represents the value of an intangible asset value, and $\Delta$PBR represents the difference between the benchmark and standalone value of the intangible asset. The geometric mean is used for the PBR population.

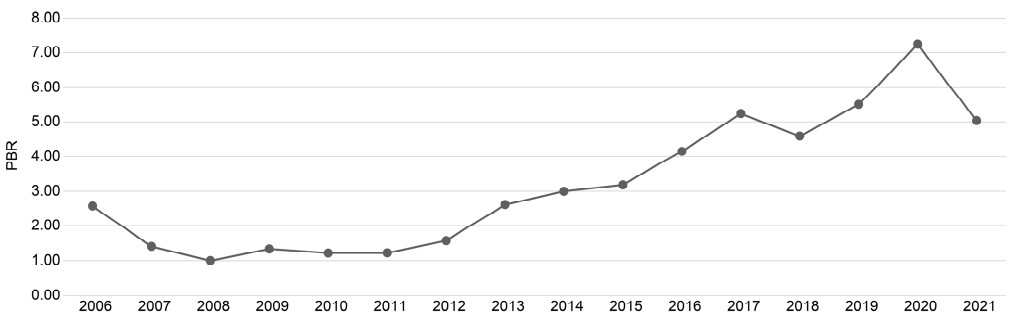

**Figure 1.** PBR time series for the benchmark of 2020's top 90 SaaS companies.

3.2.5. Standalone Value of Intangible Assets and Value Transfer between Digital Sectors

Using the greenfield method, we calculate the standalone value of the intangibles generated by each digital sector. The six digital sectors include the top 90 SaaS advanced companies in 2020; therefore, ⊿PBR represents the standalone value of SaaS intangibles. Additionally, SaaS intangible asset value is allocated to each sector based on the ratio of SaaS companies belonging to each sector.

We divide those data into two periods (2006–2012 and 2013–2021) in Assessment 2. The PBR reached its peak in 2020. TOPIXβ is the expected return for one unit of investment. Sector idiosyncrasies should be corrected using the β value, which is the expected return on investment for each sector. In the Results section, we discuss the total value of SaaS intangibles and the segregated value of human intellectual capital-oriented digital operational intangibles.

*3.3. Assessment 2: Qualitative Changes in Capital Investment and Labor Investment*
3.3.1. Purpose

To determine the value of the SaaS operation method alone, we consider the qualitative changes in capital and labor investments stemming from digital innovation and verify their consistency with the value transfer between digital sectors, which is determined in Assessment 1. Any identified discrepancies may indicate the causes of accounting distortions. Specifically, SaaS adoption may be transforming the quality of digital productivity.

Although digital intangible assets can enhance value, accurately reflecting the value transfer induced by SaaS in financial statements is difficult. Therefore, there is a need for a better understanding of the role of human intellectual capital in digital operational knowledge as a factor in the value transfer processes across the SaaS sector (Van Ark 2016).

3.3.2. Analysis Method

We verify differences in investment efficiency between cases with and without the inclusion of intangible assets. We compare the rates of return on labor investment over one unit of capital investment. By comparing differences in capital and labor investments between SaaS and non-SaaS companies, we investigate the qualitative changes in capital and labor investments.

**4. Results**
*4.1. Value Transfer Phenomenon between Digital Sectors*

In Assessment 1, we observe three distinct trends in the historical performance of the PBR in the digital sector, as shown in Figure 2. The first trend, which spans from 2006 to 2012, reveals that all six sectors remain within the range of −1 < ⊿PBR < 1. The second trend, from 2013 to 2020, highlights a general underperformance in terms of PBR across all sectors with ⊿PBR < 0 for all six sectors in 2019 and 2020. Finally, the third trend, which emerges after 2021, exhibits an improvement in PBR performance.

These results demonstrate minimal variation in IGG among the six digital sectors until 2012. However, since the onset of digitization in 2013, it has become evident that the value shifted from companies that had failed to adapt to the SaaS model to those that had successfully transitioned. This trend can be observed in the underperformance of all sectors compared with the top 90 SaaS companies benchmark in 2020. Each digital sector included both SaaS and non-SaaS companies. The more SaaS the digital sector adopts, the greater the value transfer. Additionally, value transfer occurs through the medium of unrecognized tangibles. One of the critical factors might be human intellectual capital that mediates the value transfer to the digital sector, which was previously unrecognized as a SaaS value. This means that SaaS has both perceived and unrecognized human intellectual capital. Additional research is required to understand how SaaS human intellectual capital drives this phenomenon.

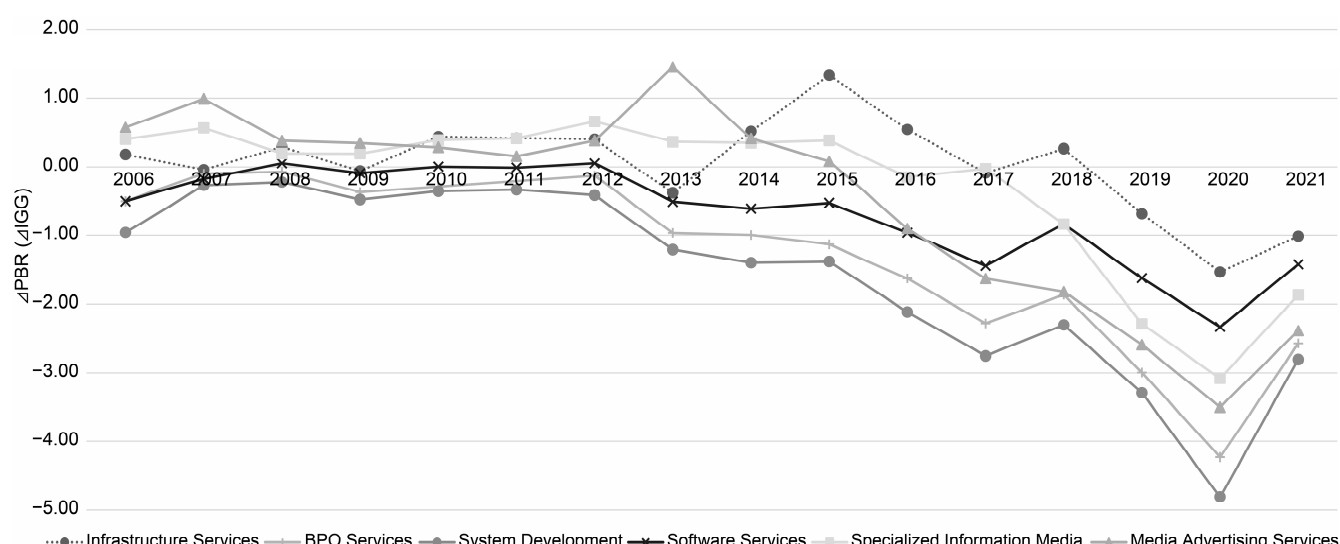

**Figure 2.** Historical ⊿PBR performance of six digital sectors.

### 4.2. Calculation of the Value Transfer to SaaS Companies

Table 2 defines the methodology used to calculate the standalone values of intangibles and rates of value transfer to each sector. Initially, the intangible asset value is determined by applying the β correction to each benchmark and sector. Subsequently, the allocation of intangible assets within each sector and SaaS ratios of all companies in the six sectors are determined. The gross amount of intangibles in each sector and the value transfer ratio relative to the benchmark are then calculated. Based on the benchmark intangible asset value of 6.23 (gross amount: JPY 3991B) and total intangible asset value of the six sectors of 3.41 (gross amount: JPY 3476B), the difference of 2.82 (gross amount: JPY 515B) represents the value transfer to SaaS companies. Further, by examining the differences in sector allocation, we determine that there are sectors with inflows of intangible asset value, namely infrastructure services and media advertising services, and sectors with outflows of intangible asset value, namely BPO services, system development, software services, and specialized information media.

**Table 2.** SaaS intangible values and value transfer rates of six digital sectors via SaaS human intellectual capital.

| | Population | TOPIX β@2020 (a) | PBR@2020 (b) | SaaS Intangible Value after Correction (c = b/a) | Intangible Allocation (d = c × Intra-sector Ratio) | Intangible Allocation (e = c × Ratio within Benchmark) | Value Transfer (Gross) (f = d − e) | Value Transfer % (f/F) |
|---|---|---|---|---|---|---|---|---|
| **Top 90 SaaS Companies** | 90 | 1.16 | 7.23 | 6.23 (C) | NA | 6.23 (100%) | 2.82 (F = C−D) | |
| **Infrastructure Services** | 41 | 1.35 | 5.70 | 4.22 | 1.24 (29.3%) | 0.83 (13.3%) | 0.41 | 14.5% |
| **BPO Services** | 58 | 1.23 | 3.00 | 2.44 | 0.13 (5.2%) | 0.20 (3.3%) | ⊿0.07 | ⊿2.48% |
| **System Development** | 195 | 1.01 | 2.42 | 2.40 | 0.20 (8.2%) | 1.11 (17.8%) | ⊿0.91 | ⊿32.3% |
| **Software Services** | 195 | 1.04 | 4.90 | 4.71 | 1.04 (22%) | 2.98 (47.8%) | ⊿1.94 | ⊿68.8% |
| **Specialized Information Media** | 166 | 1.41 | 4.15 | 2.94 | 0.15 (5.4%) | 0.62 (10%) | ⊿0.47 | ⊿16.7% |
| **Media Advertising Services** | 41 | 0.97 | 3.73 | 3.85 | 0.65 (17%) | 0.49 (7.8%) | 0.16 | 5.6% |
| **All Six Digital Sectors** | 696 | 1.15 | 3.82 | 3.32 | 3.41 (D) | NA | ⊿2.82 | ⊿100% |

### 4.3. Digital Service Productivity Investigation

Thus far, productivity discussions have been presented from the perspective of tangible assets. In Assessment 2, we present two cases. Case I delineates the prevailing context wherein the capitalization of intangible assets is proscribed. Case II delineates an idealized framework wherein both the capitalization of intangible assets on the balance sheet and their subsequent amortization are sanctioned.

To focus on digital intangible assets, digital service productivity is defined as follows:

Digital Service Productivity = (Output of Case II-Case I) ÷ (Input of Case II-Case I)

Case I: *Do not* recognize any intangible assets

Output: Value added by tangible assets = Operating Income + Personnel Expenses + Rent + Taxes and Levies in Manufacturing, Selling, and General Administration Expenses + Parent Royalty + Depreciation

Input: Tangible assets and number of employees Variable Y_Case I = Value added by tangible assets ÷ tangible assets Variable X_Case I = Value added by tangible assets ÷ number of employees

Case II: Recognize *both* tangible and intangible assets

Output: Value added by both tangible and intangible assets = Operating Income + Personnel Expenses + Rent + Taxes and Levies in Manufacturing, Selling, and General Administration Expenses + (Net) Intangible Fixed Assets + (Net) Digital Intellectual property + Depreciation + Amortization

Input: Capital assets and number of employees Variable Y_Case II = Value added by both tangible and intangible assets ÷ capital assets Variable X_Case II = Value added by both tangible and intangible assets ÷ number of employees

The two cases of *only* tangible assets and *both* tangible and intangible assets were examined as defined below. Here, the variable Y represents the rate of return on capital investment, and the variable X represents the rate of return on labor investment. Additionally, Case I does *not* recognize intangible assets, whereas Case II *does* recognize intangible assets.

$$\varDelta X = \text{variable X\_Case II-variable X\_Case I}$$

$$\varDelta Y = \text{variable Y\_Case II-variable Y\_Case I}$$

$\varDelta X$ represents qualitative human capital changes in labor investment, and $\varDelta Y$ represents qualitative changes in capital investment. Additionally, the correlation between the differences in "Growth in value-added rate per capital" and "Growth in value added per employee" is investigated. By comparing these differences, we determine the expected rate of return on labor investment per unit and the expected rate of return on capital investment per unit.

### 4.4. Capital and Labor Investment and Productivity Investigation

The six digital sectors are classified into three patterns, as shown in Table 3. As shown in Figure 3, our results indicate that some sectors are inconsistent with the results of Assessment 1. When comparing the value transfer percentages presented in Table 2 with the coefficients of X from 2013 to 2020 in Table 3, only the BPO services and specialized information media sectors are found to be inconsistent. For the remaining four sectors, the trends in the value transfer rate are consistent with the rate of return on labor investment.

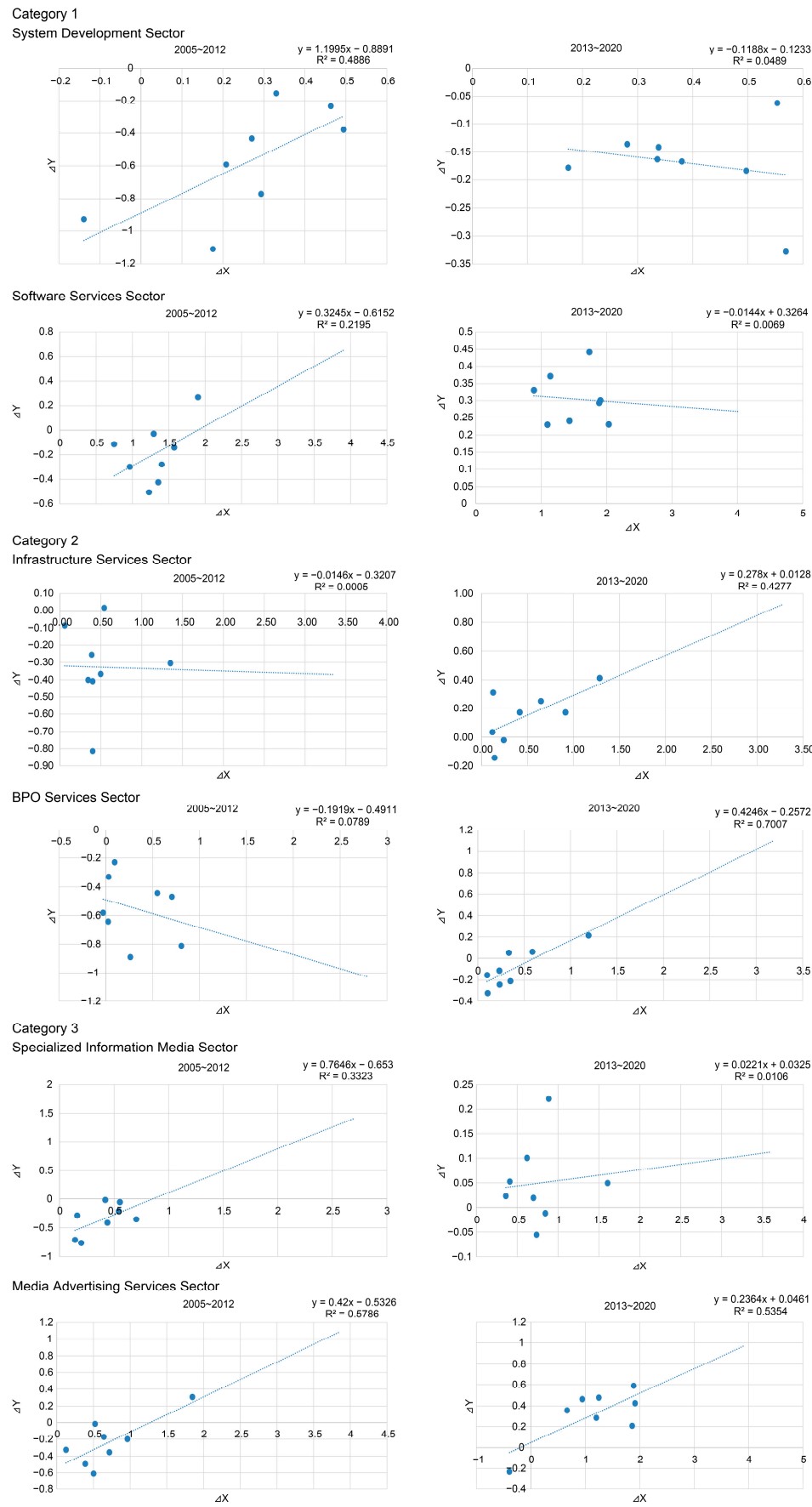

**Figure 3.** Six digital sectors' linear regression equations divided into three categories.

**Table 3.** Digital sector classification based on the coefficient of X.

| | Coefficient of X 2005–2012 | Coefficient of X 2013–2020 | Digital Sector |
|---|---|---|---|
| **Category 1** | Positive | Negative | System Development †, Software Services † |
| **Category 2** | Negative | Positive | Infrastructure Services ‡, BPO Services † |
| **Category 3** | Positive | Positive | Specialized Information Media †, Media Advertising Services ‡ |

†: IGG outflow sector in assessment 1; ‡: IGG inflow sector in assessment 1.

In Figure 3, the differences between investment productivity per unit, including and excluding investment in intangible assets, are defined along the *x*-axis. The *y*-axis represents the differences per unit of rate of return, including and excluding intangible investment.

Even within the same digital sector, labor productivity relative to capital investment differs before and after 2013. The underestimation of labor productivity versus changes in the quality of capital investment leads to the transfer of value between sectors. We constructed a rationale that is suitable for the digital revolution era by comparing labor productivity with the amount of digital investment, in-house operational innovation, and human intellectual capitalization of digital human resources. We further verified the underestimation of labor productivity versus capital investment. The value transfer rate is important for determining the percentage of the new added value generated that should be tailored to SaaS providers and users. In practice, it is logical to separate the definitions of SaaS provider and user-added value.

## 5. Discussions

### 5.1. HTVI Recognition Is Dependent on SaaS Providers and Users

The proliferation of SaaS has led to a significant shift from capital expenditure investments to cloud-based services. However, SaaS providers still require tangible assets to serve as infrastructure, whereas SaaS user companies have adopted an operating expenses approach. SaaS user companies invest only in license fees and obtain new added value as outputs from the human intellectual capital of SaaS providers, resulting in maximum output with minimum input. Regardless, SaaS users cannot record digital intangibles and their corresponding amortization, leading to accounting inconsistencies. Traditional value-added formulas do not consider the concept of value transfer adjustment, leading to an underestimation of the value of SaaS providers. Additionally, the attribution of value transfer induced by highly skilled digital operators is difficult to assess, particularly in terms of operational capabilities. Intangible assets must be included in the value-added formula to assess the productivity of digital services properly. However, balancing value transfer through operational methods is challenging for both SaaS providers and users, as shown in Figure 4. Therefore, recognizing the effective use of human intellectual capital and digital transformation of business methods is crucial for creating new added value and increasing returns on investment.

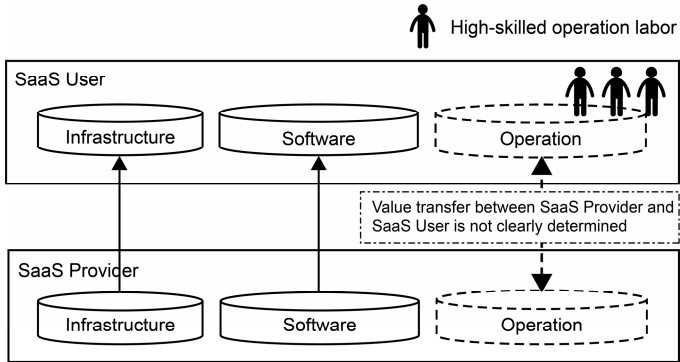

**Figure 4.** Ambiguity of human intellectual capital attribution.

*5.2. Qualitative Changes in Human Capital and Accounting Distortions*

Assessments 1 and 2 reveal that there are sectors in which digital intangibles are undervalued on the basis of insufficient accounting literacy for expressing added value. This ambiguity makes it challenging to determine the added value generated by high-skilled digital operators on either side of the SaaS provider–user relationship, resulting in underestimated intangible assets. To address this issue, "labor investment" can be redefined as "digital labor investment" and "capital investment" as "digital capital investment". This can help SaaS providers and users attribute digital human intellectual capital, such as operational knowledge, to resolve accounting distortions. In sectors such as BPO services and specialized information media, the attribution of digital operations is closely linked to SaaS users. This obscures the value transfer from SaaS providers, which can result in undervaluation because sufficient "digital capital return" for "digital labor investment" is not evaluated. Therefore, rationalizing attribution is necessary to value intangible assets appropriately. Examples of attributed HTVI include software, licenses, and goodwill amortization expenses, whereas examples of HTVI with unclear attribution include digital skills related to human intellectual capital and operational knowledge. It is necessary to consider and assess the medium through which value transfers between SaaS providers and users occur.

As shown in Figure 5, with conversion to SaaS, IGG is evaluated more highly. By properly segregating and replacing SaaS operations with internal and external operations, companies using SaaS can express new added value from their competitors. It is important to determine how SaaS operations can be effectively separated into internal and external operations to ensure the proper attribution of digital operators. Companies can benefit from an influx of intangible assets by bringing SaaS in-house using concepts such as DevOps and CI/CD. The advanced digital operational skills of employees and the quality of their work should be considered as new added value by recording them as "high-value-added digital labor". Additionally, the digitalization of labor can increase the labor unit price, create new added value, and improve digital service productivity.

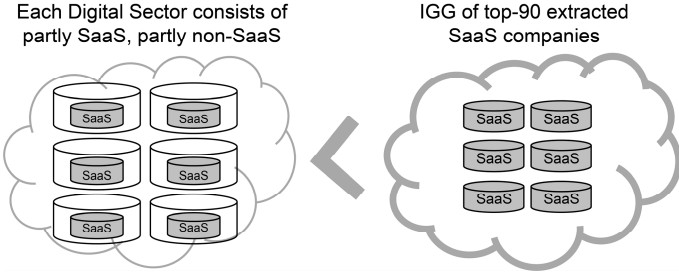

**Figure 5.** Trends of internally generated goodwill (IGG) comparison.

## 6. Conclusions

This study proposes a method for calculating the standalone value of digital intangibles using the greenfield method. We find that conversion to SaaS increases the value of intangible assets and transfer rate. Independent companies can enjoy inflows of intangible assets, and the quality of digital labor should be considered as new added value in the form of "human intellectual capital of advanced digital labor". The digitization of labor can improve the productivity of digital services, which can serve as quantitative proof of the effectiveness of basic income introduction. The following steps can be taken to assess the digital intangibles of a target company in the SaaS sector accurately.

1.  It is necessary to benchmark the PBR multiples of SaaS companies in the same sector.
2.  On the basis of this benchmark, we define a PBR multiple range for a target company's competitors.
3.  We calculate the *Δ*PBR range by subtracting the benchmark from the PBR multiple range.

4. The net assets of the target company are multiplied by the ⊿PBR range to estimate its enterprise value.

5. Finally, it is important to perform β correction and value transfer rate allocation specific to the sector to which the target company belongs to ensure accurate valuation. By following these steps, a more precise valuation of a target company in the SaaS sector can be achieved.

While the overall value of SaaS can be calculated, it is not feasible to identify the exact factor of qualitative human changes taking place. A forthcoming objective is to break down human intellectual capital and refer to changes in critical factors. For instance, measurable factors, such as the ratio of operation functions converted to SaaS, the ratio of in-house production by DevOps and CI/CD, and the ratio of high-value-added digital labor, can be evaluated. Subsequent research should elucidate digital service productivity at an in-depth level and identify the human intellectual capital factors with the highest correlation.

**Author Contributions:** Conceptualization, S.S.; methodology, S.S.; validation, T.F.; formal analysis, S.S.; data curation, S.S.; writing—review and editing, T.F.; supervision, T.F. All authors have read and agreed to the published version of the manuscript.

**Funding:** This research received no external funding.

**Institutional Review Board Statement:** Not applicable.

**Informed Consent Statement:** Not applicable.

**Data Availability Statement:** The datasets used and analyzed during the current study are available from the corresponding author upon reasonable request.

**Conflicts of Interest:** The authors declare no conflicts of interest.

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
