# Peer review of "Standalone Valuation Method for Software-as-a-Service Operational Knowledge Derived from Human Intellectual Capital Qualitative Changes"

_admsci, doi:10.3390/admsci14040071_

Round 1

Reviewer 1 Report

Comments and Suggestions for Authors

How is your way of assessment specifically better than those of the previous studies concerned? More generally what is your main contribution to the literature? Should be mentioned in Section 1 more clearly.

Ll.53-58: Terms such as DevOps and CI/CD needs to be more elaborated so that you can more clearly link them to the main discussion.

Ll.104-107: This paragraph should be moved to the end of the first section.  The description on the correct Section 2 has to be added and the remaining section numbers need to be revised.

Most of Section 3 should be moved to Section 4. Moreover, All the Section 4 should be to Section 5, as it does not contain any substantial analysis like Section 3, rather that will be used as a basis for interpreting the results of BPO services and specialized information media.

l.137: Why did you select the top 90 as the benchmark? Why not 80, 100 or else? You should show the criteria.

Did all the top 90 companies start SaaS exactly in 2013? Some might have started more recently. Had all the non-top 90 companies in the six sectors implemented no SaaS by 2021? Some might have implemented by then.  If the answers are yes or no, but somehow negligible, no problem. Otherwise, Assesment 1 must be biased. More complicated if there have been any companies that once started SaaS but quitted by 2021?

More seriously, the assessment assumed the contributions of non-SaaS operations are same or at least not significantly different between the top 90 and the others. This is likely to be too a strong assumption.

Based on the potential biases raised by the reasons above, it seems to be too bold to argue that all the delta PBR should be attributed to the operational knowledge derived from human intellectual capital.

3.2.7: You can calculate the monetary term of intangible asset differences between the top 90 and the others. Rather than showing only 2.82 based on PBR, it is more understandable by the broader scope of readers.

Figure 3: Are the 2013-2020 results of system development sector and software services sector statistically significant? If no, you should not show "-" in table 3. The interpretations do not seem to require overall adjustment, but need to reconsider.

Reviewer 2 Report

Comments and Suggestions for Authors

The article is very interesting, and I have read it with pleasure. To develop a quantifiable valuation method for intangible assets, as human resources are very important intellectual capital in an organisation, can bring accounting further. But there are still some questions to answer.

I miss specific definitions of the most important constructs in the article, and references belonging to it.

Specific comments

In line 22 the authors refer to Germany (‘ The advent of the digital revolution in Germany in 2013 resulted in the emergence of various forms of digital intangibles, particularly in relation to the attribution of employee digital literacy, which is also known as digital operational knowledge.’) So, it suggests that the study would continue referring to and being conducted in Germany. But in line 110 it becomes clear that the study is conducted in Japan (‘Assessment 1 examined the value 110 transfer phenomenon across digital sectors in the Japanese market.’). How does this fit? Was the development of the digital revolution in Japan the same as in Germany?

Line 113. Please explain why internal generated goodwill (after defining it) is such a good evaluation index for intangible assets.

Line 222 and further, in section 3.3.3 Digital service productivity the authors introduce two cases. Could these casus be introduced more?

Round 2

Reviewer 1 Report

Comments and Suggestions for Authors

Thank you for your revision. Since I found some more room for improvement, I made comments below to your response. Hope that they will be beneficial to you so that you will be able to publish the manuscript after another round of revision.

Reviewers initial comment

l.137: Why did you select the top 90 as the benchmark? Why not 80, 100 or else? You should show the criteria.

※Author’s comment

In 2020, a SaaS bubble occurred in Japan. We extracted the top 100 sales and obtained data from individual companies going back from 2005 up to 2021. In this process, we omitted 10 companies due to missing data, resulting in a total of 90 companies.

Reviewer's comment this time

I don't mean you have to revise further, but just for your reference for your future studies, whichever is top 90 or top 100 does not matter. In order to make a more rational way of selecting "good performers", for example, companies achieving performance equal to and better than "mean + one standard deviation" seem to be a good option or more simply the first quantile companies are still acceptable.

Reviewer's initial comment

Did all the top 90 companies start SaaS exactly in 2013? Some might have started more recently. Had all the non-top 90 companies in the six sectors implemented no SaaS by 2021? Some might have implemented by then. If the answers are yes or no, but somehow negligible, no problem. Otherwise, Assesment 1 must be biased. More complicated if there have been any companies that once started SaaS but quitted by 2021?

※Author’s comment Thank you for pointing out this very important issue. In 2010, ASP (Application Service Provider) was mainly used. Companies had their own on-premises infrastructure and there was no distinction between SaaS and non-SaaS. Figure 2 of Assessment 1 shows the development of cloud infrastructure (e.g., Azure, AWS, GCP) since 2013. We can see how SaaS gradually outperformed non-SaaS from the perspective of IGG. If some companies started and quit SaaS between 2013 and 2021, we presume this would not affect the overall trend because the population is sufficiently large.

Reviewer's comment this time

I am afraid your reply is not persuasive enough, particularly regarding how your presumption can be justified. You should provide evidence your presumption is not too strong. I am sure this point is very critical, because your main argument is not justifiable. If you will not find any supporting evidence, at least you should refer to it as a limitation of the study.

Reviewer's initial comment

More seriously, the assessment assumed the contributions of non-SaaS operations are same or at least not significantly different between the top 90 and the others. This is likely to be too a strong assumption.

※Author’s comment

Thank you for your thoughtful comments. The OECD defines HTVI (lines 330–334) as software, licenses, and goodwill intangible assets that clearly belong to a company. The knowledge and know-how of digital operations of employees (natural persons) are cited as an example of HTVI. For this reason, this study found that there were no qualitative differences between the operations of 90 SaaS companies and the other non-SaaS companies. Instead, we focused on how HTVIs that do not belong to companies have been quantitatively evaluated historically. The gross IGG of the six sectors including the 90 SaaS companies should be larger than the IGG of the 90 SaaS companies alone. However, we show that the opposite occurs in the real world, as there are some HTVI materials that intermediate the value transfer between SaaS and non-SaaS (see Figure 5).

Reviewer's comment this time

My point is as same as that of the previous comment such that the top 90 companies may have non-SaaS operations while the others may have SaaS operations. Allow me to repeat, but under this condition, we cannot attribute the performance difference between the two groups to the superiority of SaaS operations of the top 90 companies.

Even though this way is likely to be out of your scope of intended contribution, but I can agree to a more modest explanation without hesitation, such that; the performance difference is found between the top 90 companies and the others. Judging from the observation of the different business scopes between the two groups, SaaS may be one of the main reasons for this. Although you may feel my idea is not so different from yours, this cautiousness is required for academic argument, in my understanding.

As to your replies to my other comments, your revisions are acceptable. I appreciate all your efforts.
